# A Nationwide Survey of Dementia Prevalence in Long-Term Care Facilities in Taiwan

**DOI:** 10.3390/jcm11061554

**Published:** 2022-03-11

**Authors:** Yi-Hui Kao, Chih-Cheng Hsu, Yuan-Han Yang

**Affiliations:** 1Department of Medical Education and Research, National Taiwan University Hospital Yun-Lin Branch, Douliu 640, Taiwan; blueggobi@gmail.com; 2Graduate Institute of Anatomy and Cell Biology, National Taiwan University College of Medicine, Taipei 100, Taiwan; 3Department of Neurology, National Taiwan University Hospital, Taipei 100, Taiwan; 4National Center for Geriatrics and Welfare Research, National Health Research Institutes, Zhunan 350, Taiwan; 5Institute of Population Health Sciences, National Health Research Institutes, Zhunan 350, Taiwan; 6Department of Health Services Administration, China Medical University, Taichung 404, Taiwan; 7Department of Family Medicine, Min-Sheng General Hospital, Taoyuan 330, Taiwan; 8Department of Neurology, Kaohsiung Municipal Ta-Tung Hospital, Kaohsiung City 801, Taiwan; 9Department of Neurology, Kaohsiung Medical University Hospital, Kaohsiung Medical University, Kaohsiung City 807, Taiwan; 10School of Post-Baccalaureate Medicine, Colleague of Medicine, Kaohsiung Medical University, Kaohsiung City 807, Taiwan; 11Neuroscience Research Center, Kaohsiung Medical University, Kaohsiung City 807, Taiwan

**Keywords:** aging, Alzheimer’s disease, dementia, dementia prevalence, hypertension, institution, long-term care

## Abstract

Background: As the average life expectancy of global citizens has increased, the prevalence of dementia has increased rapidly. The number of patients with dementia has increased by 6.7 times, reaching 300,000 in the past three decades in Taiwan. To realize the latest actual situation, the need for institutional care for elderly patients with dementia, and also a reference basis for government agencies to formulate dementia-related care policies, we investigated the institutional prevalence of dementia. Methods: We randomly sampled 299 out of the 1607 registered long-term care facilities including senior citizens’ institutions, nursing homes, and veteran homes in every administrative region of Taiwan. Then, a two-phase survey including MMSE screening, CDR, and clinical confirmation was conducted on each subject from 2019 to 2020. Results: Among 5753 enrolled subjects, 4765 from 266 facilities completed the examinations with a response rate of 82.8%. A total of 4150 subjects were diagnosed with dementia, 7.4% of whom had very mild dementia. The prevalence of all-cause dementia, including very mild dementia, was 87.1% in all facilities, 87.4% in senior citizens’ institutions, 87.1% in nursing homes, and 83.3% in veteran homes. Advanced age, low education, hypertension, Parkinsonism, respiratory disease, stroke, and intractable epilepsy were associated with dementia risk. Conclusions: We show that in an aged society, the prevalence of all-cause dementia in long-term care institutions can be as high as 87.1%. This study was completed before the outbreak of COVID-19 and provides a precious hallmark for future epidemiological research. We recommend that the long-term care policy in an aged society needs to take into account the increasing high prevalence of dementia in the institution.

## 1. Introduction

The average life expectancy of citizens in most countries has risen gradually with the advancement of medical care. Many countries face varying degrees of social ageing issues. Old age is the leading non-modifiable risk factor for dementia including both Alzheimer’s disease and vascular dementia [1,2]. Dementia not only leads to cognitive decline but also psychiatric and behavior problems [3]. The burden of medical costs, care manpower, and psychological pressure seriously affect family members with dementia and society [4,5,6,7]. 

Long-term care institutions are a common way of caring for the elderly with dementia. The prevalence of dementia in long-term institutions varies greatly, ranging from 16.1% to 85.2% according to factors such as the country, aging degree, investigation timing, and research method [8,9,10,11,12,13]. Besides, culture, religion, race, urbanization, welfare policy, insurance, dependency ratio, and many other factors influence the prevalence of dementia in the institution [14,15]. In addition, many studies focus on specific types of institution or certain geographic areas with a relatively small sample size. Nearly 70% of studies on dementia prevalence in long-term care facilities have been performed in Europe [16]. Only limited studies have been conducted in Asia, the Americas, Africa, or Eurasia (Appendix C). Chen et al. [17] in 2007 reported that 48% of residents of long-term care wards in Taiwan had dementia. Guo [18] and Xu [19] et al. subsequently reported 36.7% and 44.5% in China. Therefore, a study in Asia that comprehensively covers all institutions and minimizes sampling bias is needed to provide a holistic perspective on the topic.

Among factors associated with prevalence of dementia, the degree of social aging is an important issue and it significantly affects how people choose the way to take care of elders with dementia. The World Health Organization defines 7%, 14%, and 20% of the total population as over 65 years old, which includes aging, aged, and super-aged societies, respectively. After crossing the threshold of an aging society in 1993, Taiwan quickly reached an aged society in 2018. 

There were 328.2 million elderly people worldwide in 1990, and by 2020 this number had more than doubled to 727.6 million. Among them, more than 50 million people are now suffering from dementia. During the same period, in 1990, there were an estimated 45,000 dementia patients in Taiwan [20,21]. By 2020, this number increased 6.7 times to reach 303,271 out of a population of 23 million. The prevalence of dementia in long-term care institutions is dynamic and up-to-date research is crucial for public health policy. Due to the rapid aging of society in recent years and the massive increase in the population of dementia patients, the National Health Research Institutes in Taiwan conducted an epidemiological survey on the prevalence of dementia in long-term care facilities. The aims of this study were to realize the latest actual situation, the need for institutional care for elderly patients with dementia, and also a reference basis for government agencies to formulate dementia-related care policies. 

## 2. Materials and Methods

This was a cross-sectional study including 6549 subjects from all categories of long-term care units in Taiwan. Experiments with a two-stage random sampling design were conducted between July 2019 and February 2020. All administrative regions were included in this national study.

### 2.1. Type of Long-Term Care Facilities

There are 3 categories of long-term care units in Taiwan. First, senior citizens’ institutions include residential houses for healthy elders living independently and assisted living facilities for people who need some support in activities of daily living. Second, nursing homes accommodate people with serious illnesses or those dependent on medical care. Third, veteran homes mainly take care of retired soldiers from the national army who are old or sick. This study included the above three types of long-term care institutions. 

### 2.2. Estimation of Sample Size

This epidemiological investigation was designed by the Taiwan National Health Research Institutes. There are 1607 long-term care units registered in Taiwan long-term care of the Ministry of Health and Welfare. According to the estimation formula proposed by Daniel and Cross [22], we estimated that 6549 subjects would be sampled from 22 administrative regions including Taiwan island and outlying islands including Penghu, Kinmen, and Lianjiang. 

If the ratio of the sample size to the population size is greater than 0.05, then a limited population correction factor needs to be considered. The formula is as follows: 

n = sample size

N = the number of populations; 

P = proportion for population;

d = precision.
$$n=\frac{NZ^{2}P\left(1-P\right)}{d^{2}\left(N-1\right)+Z^{2}P\left(1-P\right)}$$

According to previous research, regardless of the type of institution, the prevalence rate of institutional dementia is estimated to be 45.67%. Suppose the precision of the prevalence rate is 5%. 

### 2.3. Sampling Method: Two-Stage Random Sampling

We stratified randomized sampling by 22 administrative regions and followed the principle of withdrawing and not returning. The probability of an institution being sampled should reflect the number of residents of the institution. We took 100 people as the sampling unit. Institutions with fewer than 100 residents occupied one lottery ticket; institutions with 100–200 residents occupied two lottery tickets, and so on. As a result, a total of 299 institutions were selected, including 164 nursing homes, 125 nursing homes, and 10 veteran homes.

Then, 6549 subjects were randomly selected from the list of residents of the above-mentioned institutions (Appendix A). To reflect the number of people in the three types of long-term care institutions, the estimated sample number of each administrative region was allocated to the survey sample number according to the proportion of the number of people accommodated by the types of institutions. 

If the institution sampled in the first stage or the residents sampled in the second stage could not cooperate with the investigation, a substitute sample would be drawn according to the principle of random sampling. 

### 2.4. Two-Phase Survey of Subjects

We reviewed the medical profile of every subject (Figure 1). If the subject was confirmed to have dementia, we recorded the diagnosis, the severity of the disease, and filled out the questionnaire. In the remaining cases, we conducted a dementia assessment with a Two-phase Survey. 

In the first phase, well-trained evaluators visited the intuitions between July and November 2019. All sampled residents received the Taiwanese Mental State Examination, a version of the Mini-Mental State Examination (MMSE) [23,24,25], assessment for activities of daily living (ADL), and instrumental activities of daily living (IADL) [26]. Barthel Index [27] was used for evaluation of ADL. IADL was assessed according to Lawton and Brody’s design [26]. Subjects who self-reported cognitive decline, MMSE scores below the critical value, or were difficult to evaluate were included in the second phase of the evaluation. The critical value was defined as an MMSE score less than 25 if the subject was literate or less than 14 if not literate. 

Then, experienced neurologists and psychiatrists visited the subjects, made a diagnosis and conducted a Clinical Dementia Rating (CDR) [28] between December 2019 and February 2020. The assessment of the subject’s CDR was carried out with the assistance of the main caregiver of the institution. All neurologists and psychiatrists participated in the education and training organized by the society before the evaluation. A CDR score equal to 0.5 points was considered very mild dementia (VMD) [29] and a score greater than 0.5 points was diagnosed as dementia. 

### 2.5. Statistical Analysis

The data were expressed as mean (standard deviation) and number (%) for continuous and categorical variables, respectively. The group difference results were examined using with the Kruskal–Wallis t-test and chi-squared test for continuous and categorical variables, respectively. We assessed weighted prevalence of dementia by using SUDAAN software (version 11.0.1, RTI International, Research Triangle Park, NC, USA) to account for sampling effects. The rest of statistical analyses in this study were performed by SAS (version 9.4 for Windows; SAS Institute, Inc., Cary, NC, USA).

This study was reviewed and approved by the Medical Research Ethics Committee of the National Health Research Institutes, number EC1080502. All subjects or their family members signed an informed consent form.

## 3. Results

This epidemiological study was conducted between July 2019 and February 2020. We completed sampling of 266 institutions including 143 senior citizens’ institutions, 113 nursing homes, and 10 veteran homes from 22 administrative regions including Taiwan Island and outlying islands including Penghu, Kinmen, and Lianjiang. The averaged institutional response rate was 89%, with 87% senior citizens’ institutions, 90% nursing homes, and 100% veteran homes, respectively.

### 3.1. Demographic Data

Among 5753 enrolled subjects (Appendix B), 4765 completed the 2-phase examination with an 82.8% response rate. The reasons for failure to complete the tests included closed institutions, discharge from the institution, and refusal for interview. The demographic results are shown in Table 1.

The sexratio of all enrolled subjects was almost equal except for more man in veteran homes. The mean age was 76.98 ± 13.39 and most respondents were illiterate. People living in the veteran homes were oldest followed by senior citizens’ institutions and nursing homes (Bonferroni post hoc test, *p* < 0.0001). Residents in the veteran homes also had more education years than those who lived in senior citizens’ institutions and nursing homes (*p* < 0.0001). 

### 3.2. Dementia Prevalence

The prevalence of all-cause dementia, including very mild dementia was 87.1% in all facilities, 87.4% in senior citizens’ institutions, 87.1% in nursing homes, and 83.3% in veteran homes (Table 1). There was no significant difference (*p* = 0.2116) in the prevalence of dementia among the three institutions, all exceeding 80%. The weighted prevalence adjusted by SUDAAN software was 88% (Table 2). Dementia prevalence in women was slightly higher than in men. More than 90% of institutional residents over 75 have dementia. The mean CDR of all residents with dementia or very mild dementia was 2.37 ± 0.89. It was highest in the elderly staying at nursing homes 2.42 ± 0.86, followed by senior citizens’ institutions 2.37 ± 0.89, and was lowest in veteran homes 1.76 ± 0.95 (*p* < 0.0001). A total of 61.6% of all dementia residents were diagnosed at a severe stage. The mean MMSE of all residents with dementia or mild cognitive impairment was 17.18 ± 6.82.

### 3.3. Comparison between People with Dementia and Normal Elderly

The elderly without cognition decline in all types of institution were younger than the elderly with dementia or VMD (*p* < 0.0001) (Table 3). There were more men in the elderly without cognition decline (*p* < 0.0001). Besides, the elderly without cognition decline had more education years than the cognition decline group (*p* < 0.0001). Most elderly with cognition decline were not literate while more than a quarter of normal elderly received at least 10 years of education. Elderly with cognition decline had poorer ADL and IADL than normal elderly (all *p* < 0.001) (Table 3). The residents in veteran homes had the best ADL and IADL while those who stayed at nursing homes had the worst (all *p* < 0.001). The elderly with dementia are significantly older than the normal elderly by more than 5 years. In the group with impaired cognitive function, more elderly people have hypertension, respiratory diseases, Parkinsonism, stroke, and refractory epilepsy (Table 4). There was no difference in diabetes, skeletal disease, impaired vision, coronary artery disease, cardiac arrhythmia, cancer, digestion disease, and psychiatric disease.

## 4. Discussion

Compared with a previous study [17] in Taiwan 15 years ago, the prevalence of dementia in long-term care units increased dramatically from 45.7% (26.8–64.5%, depending on the type of institution) to 87.1%. Among them, 7.4% of residents were diagnosed with VMD. In general, this means that about 85% of institutional residents have varying degrees of cognitive dysfunction. We expect that the prevalence of dementia in long-term care units could increase but the result far exceed expectations. Besides, the mean age of this study (76.98 ± 13.39) was even smaller than that of the previous study (79.4 ± 7.2) [17]. Traditionally, Taiwanese tend to take care of their elders at home. Sending parents to an institution for care may be considered unfilial, so this is usually not the first choice. However, the prevalence of dementia in institutions is still rising sharply, regardless of resident age. We speculate that there may be some possible explanations for these findings.

First, age as the main and inevitable risk factor for dementia has impacted greatly on incidence. When the average life expectancy increases, the incidence of dementia rises accordingly [7]. Over the past three decades, the prevalence of dementia nationwide has increased 4.5 times from 1.7 to 8.04% in Taiwan [2,20,30,31,32,33]. In 2004, there were an estimated 90,000 dementia patients in 23 million populations. It took only 16 years for the number to more than triple to 291,000 without much increase in the total population. Alzheimer’s disease and vascular dementia were most common causes of dementia [2,34]. The rapidly increasing number of patients with dementia makes the society difficult to cope with.

Second, Alzheimer’s disease as the most common dementia is a neurodegenerative disease and progresses slowly. Advanced medical treatment [35] and proper nursing care may increase survival from dementia diagnosis, and therefore also lengthen the patient’s incapacity time after illness. Besides, with the implementation of the long-term care policy, people are more aware of dementia and patients are diagnosed earlier. Dementia survival time is negatively associated with age at diagnosis [36]. The prevalence is based on the incidence of the disease and duration of illness. With the increase in the incidence of dementia and the survival time of dementia, the prevalence has increased sensibly.

Third, this study randomized sampled subjects from all 22 administrative regions across the country and proportionally distributed subjects in all kinds of institutions. Compared with the previous study [17], some counties with a degree of aging higher than the national average, such as Miaoli and Yilan, were also included in this study. Yilan County is located in the eastern part of Taiwan, which has the highest prevalence of dementia in Taiwan [33]. Since counties with older age and higher prevalence of dementia were included, the prevalence of this study also increased.

Finally, the difficulty of caring for people with dementia is well known. Taiwan reached an aging society in September 1993 and kept going at an extremely rapid rate. It took less than 25 years for people aged 65 years to double and the country entered an aged society in March 2018, two years earlier than expected. Even more amazing is that it is estimated that it will only take 7 years to enter the super-aged society in 2025. As an aged society, the old age dependency ratio increased rapidly from 10.48 in 1993 to 20.07 in 2018. Meanwhile, the aging index increased 4 times from 28.2 to 112.6 according to the Taiwan Ministry of the Interior. The number of members per household also decreased rapidly. Change in family structure leads to fewer caregivers in the family. Young people are the main source of income for the family. It is not economical if they take care of their elders at home. Besides, caring for patients with dementia is physically and labor intensive and people often cannot take care of the patient alone.

Probably based on the above factors and study design, the prevalence of dementia in institutions varies greatly in various regions of the world. Reports of institutional dementia prevalence were approximately 49.9% in the Jerusalem area [12], 56.9% in Canada [11], 62–88% in the United Kingdom [9,10], 82.8% in Norway [37], 85.2% in Austria, and 53.0% in the Czech Republic [8] (Appendix C). In the United States, 40% of assisted living facility residents [38] and 50% of nursing home residents [39] had dementia in 2014. Our finding was similar or slightly higher than that in the United Kingdom, Norway, and Austria.

The study also pointed out an important VMD group that has the opportunity to be treated [29]. If we go with the flow, VMD progresses to dementia at a rate of 10–15% every year [40]. Among all cognition decline residents, veteran homes host most VMD patients (17.8%), follow by senior citizens’ institutions (7.8%). These findings hint that not only the prevalence of dementia, but also the severity of it, varies among various institutions. Timely interventions including cognition stimulation therapy [41] are more valuable in specific institutes.

The strength of this study is mainly related to its large sample size and to it completely including 22 administrative regions across the country and sampling subjects from senior citizens’ institutions, nursing homes, and veteran homes according to the population ratio of every county. Compared with the previous report [17], this study included more than three times as many subjects (4765 vs. 1308). In addition, there are three more counties than the previous study [17] including Miaoli County, Yilan County, and Lianjiang County. Moreover, veteran homes mainly accommodating male residents were first included. This study can fully present the most complete state of the residents of the institution.

There are some limitations in this survey. First, our trained evaluators and physicians interviewed residents in different institutes by history taking, MMSE, and CDR. We lacked laboratory reports, brain image studies, and other evaluation scores. Therefore, we could not offer information about subtypes of dementia. Besides, 988 subjects were lost with an 82.8% response rate. The reasons for failure to complete the tests included closed institutions, discharge from the institution, traffic distance, and refusal for interview.

This study was conducted between July 2019 and February 2020 and revealed authentic epidemiological findings from a world without COVID-19. Pandemic infectious disease inevitably impacts on vulnerable elderly, especially those diagnosed with dementia or staying at institutions. Patients with dementia cannot stand wearing a mask for a long time. It is even more difficult for them to maintain social distancing. What is worse is that once the epidemic begins in the institution, the result is often out of control. Patients with various degrees of dementia may be vulnerable groups with high mortality rates under the epidemic disease. In some condition, their family members are forced to isolate the patient at home strictly, but that may increase the physical and mental stress on both the patient and their family. Therefore, the prevalence of dementia may be affected as the virus spreads.

In conclusion, we have shown that in a rapidly aged society, the prevalence of all-cause dementia in long-term care institutions can be as high as 87.1%. The dynamics of dementia prevalence in long-term care units reminds us of the importance of timely health policy and social resources. This study was completed before the outbreak of COVID-19 in Taiwan and could provide a precious hallmark for future epidemiological research.

## Figures and Tables

**Figure 1 jcm-11-01554-f001:**
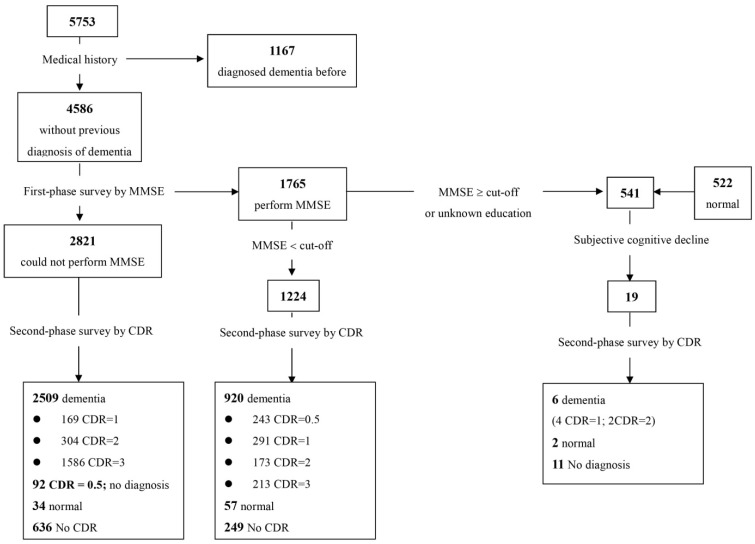
Schematic flow diagram of the sample size.

**Table 1 jcm-11-01554-t001:** Institutional basic profiles.

		Total		Senior Citizens’ Institutions		Nursing Homes		Veteran Homes		*p*-Value
Number	(*n*)	4765		2504		2033		228		
Gender	Male (*n*, %)	2308	48.4%	1066	42.6%	1025	50.4%	217	95.2%	0.0001
	Female (*n*, %)	2457	51.6%	1438	57.4%	1008	49.6%	11	4.8%	
Age	(years, mean ± SD)	76.98 ± 13.39		79.35 ± 10.74		73.18 ± 15.39		84.79 ± 11.03		0.0001
Education		(*n* = 4633)		(*n* = 2431)		(*n* = 1982)		(*n* = 220)		0.0001
	Illiterate (*n*, %)	1757	37.9%	1087	44.7%	648	32.7%	22	10.0%	
	Literate, less than 6 years (*n*, %)	1613	34.8%	848	34.9%	681	34.4%	84	38.2%	
	7–9 years (*n*, %)	468	10.1%	187	7.7%	263	13.3%	18	8.2%	
	More than 10 years (*n*, %)	789	17.0%	308	12.7%	388	19.6%	93	42.3%	
	Other * (*n*, %)	4	0.1%	1	0.04%	0	0.0%	3	1.4%	
Dementia	(*n*)	4150		2189		1771		190		0.2116
	(%, 95% CI)	87.1 (86.1–88.0)		87.4 (86.1–88.7)		87.1 (85.6–88.5)		83.3 (77.9–87.9)		
MMSE	(mean ± SD)	17.16 ± 6.81 (*n* = 1798)		16.71 ± 6.74 (*n* = 956)		17.50 ± 6.87 (*n* = 698)		18.44 ± 6.75 (*n* = 144)		0.0018
CDR	(mean ± SD)	2.37 ± 0.89 (*n* = 3919)		2.37 ± 0.89 (*n* = 2027)		2.42 ± 0.86 (*n* = 1723)		1.76 ± 0.95 (*n* = 169)		0.0001
	CDR 0.5 (mean ± SD)	291	7.4%	159	7.8%	102	5.9%	30	17.8%	
	CDR 1 (mean ± SD)	546	13.9%	270	13.3%	231	13.4%	45	26.6%	
	CDR 2 (mean ± SD)	668	17.1%	339	16.7%	285	16.5%	44	26.0%	
	CDR 3 (mean ± SD)	2414	61.6%	1259	62.1%	1105	64.1%	50	29.6%	
ADL score	(mean ± SD)	24.99 ± 31.94		25.59 ± 32.79		20.99 ± 28.83		54.10 ± 33.46		0.0001
IADL score	(mean ± SD)	1.00 ± 1.65		1.05 ± 1.72		0.81 ± 1.44		2.02 ± 2.09		0.0001

Clinical Dementia Rating (CDR), Mini-Mental State Examination (MMSE), activities of daily living (ADL), and instrumental activities of daily living (IADL). * Foreign language education, military school.

**Table 2 jcm-11-01554-t002:** Institutional dementia prevalence: crude prevalence and SUDAAN-weighted prevalence.

		Institutional Resident (*n*)	Dementia Patient (*n*)	Crude Prevalence (%, 95% CI)	SUDAAN-Weighted Prevalence (%, 95% CI) #
Total		4765	4150	87.1 (86.1–88.0)	88.0 (86.4–89.4)
Gender	Male	2308	1967	85.2 (83.8–86.7)	86.8 (84.7–88.7)
	Female	2457	2183	88.9 (87.5–90.1)	89.1 (87.0–90.9)
Age (years)	≤65	879	699	79.5 (76.7–82.1)	78.8 (74.3–82.7)
	>65	3886	3451	88.8 (87.8–89.8)	90.1 (88.5–91.5)
	≤75	1759	1435	81.6 (79.7–83.4)	81.7 (78.6–84.4)
	>75	3006	2715	90.3 (89.2–91.4)	91.7 (90.2–92.9)

# SUDAAN is a statistical software package.

**Table 3 jcm-11-01554-t003:** Comparison between people with dementia and normal elderly.

		Total		Elderly with Dementia		Normal Elderly		*p*-Value
Number	(*n*)	4765		4150		615		
Gender	Male (*n*, %)	2308	48.4%	1967	47.4%	341	55.5%	0.0002 *
	Female (*n*, %)	2457	51.6%	2183	52.6%	274	44.6%	
Age	(years, mean ± SD)	76.98 ± 13.39		77.67 ± 13.17		72.27 ± 13.96		<0.0001 *
Education		(*n* = 4633)		(*n* = 4023)		(*n* = 610)		<0.0001 *
	Illiterate (*n*, %)	1757	37.9%	1537	38.2%	220	36.1%	
	Literate, less than 6 years (*n*, %)	1613	34.8%	1482	36.9%	131	21.5%	
	7–9 years (*n*, %)	468	10.1%	382	9.5%	86	14.1%	
	More than 10 years (*n*, %)	789	17.0%	616	15.3%	173	28.4%	
	Other * (*n*, %)	4	0.1%	4	0.1%	0	0.0%	
MMSE	(mean ± SD)	17.16 ± 6.81 (*n* = 1798)		14.38 ± 5.54 (*n* = 1217)		22.97 ± 5.40 (*n* = 581)		<0.0001 *
CDR	(mean ± SD)	2.37 ± 0.89 (*n* = 3919)		2.37 ± 0.89 (*n* = 3917)		0.50 ± 0 (*n* = 2)		0.0092 *
	CDR 0.5 (*n*, %)	291	7.4%	289	7.4%	2	100%	
	CDR 1 (*n*, %)	546	13.9%	546	13.9%	0	0%	
	CDR 2 (*n*, %)	668	17.1%	668	17.1%	0	0%	
	CDR 3 (*n*, %)	2414	61.6%	2414	61.6%	0	0%	
ADL score	(mean ± SD)	24.99 ± 31.94		20.34 ± 29.15		56.41 ± 32.25		<0.0001 *
IADL score	(mean ± SD)	1.00 ± 1.65		0.69 ± 1.31		3.05 ± 2.15		<0.0001 *

Clinical Dementia Rating (CDR), Mini-Mental State Examination (MMSE), activities of daily living (ADL), and instrumental activities of daily living (IADL). * Foreign language education, military school.

**Table 4 jcm-11-01554-t004:** Comorbidities.

Disease	Total		Elderly with Dementia		Normal Elderly		*p*-Value ^a^
(*n*)	4764		4149		615		
Hypertension (*n*, %)	2812	59.0%	2472	59.6%	340	55.3%	0.0432 *
Respiratory diseases (*n*, %)	581	12.2%	534	12.9%	47	7.6%	0.0002 **
Parkinsonism (*n*, %)	336	7.1%	308	7.4%	28	4.6%	0.0095 **
DM (*n*, %)	1399	29.4%	1204	29.0%	195	31.7%	0.1719
Skeletal system disease (*n*, %)	366	7.7%	309	7.5%	57	9.3%	0.1136
Visual system disease (*n*, %)	170	3.6%	145	3.5%	25	4.1%	0.4768
Stroke (*n*, %)	1461	30.7%	1312	31.6%	149	24.2%	0.0002 **
Coronary artery disease (*n*, %)	638	13.4%	553	13.3%	85	13.8%	0.7378
Atrial fibrillation or other rhythm disorders (*n*, %)	118	2.5%	103	2.5%	15	2.4%	0.9484
Cancer (*n*, %)	127	2.7%	106	2.6%	21	3.4%	0.2167
Digestive system diseases (*n*, %)	684	14.4%	600	14.5%	84	13.7%	0.5962
Psychiatric disease (*n*, %)	697	14.6%	610	14.7%	87	14.2%	0.7158
Refractory epilepsy (*n*, %)	164	3.4%	153	3.7%	11	1.8%	0.0159 *

^a^ Chi-squared test was used for category variable. * *p* < 0.05; ** *p* < 0.01.

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
