# Peer review of "A Nationwide Survey of Dementia Prevalence in Long-Term Care Facilities in Taiwan"

_jcm, 2022, doi:10.3390/jcm11061554_

Round 1

Reviewer 1 Report

This is an interesting study which investigates the dementia prevalence in long-term care facilities in Taiwan based on a relatively large sample size. While, the author needs to clarify below points:

  1. In the introduction part, authors didn’t provide sufficient information regarding the research questions and research significance;
  2. Statistical analysis lacks sufficient information including the data analysis software, significant level, what are dependent/ independent variables in the linear model, in what situation Chi-Squared Test was used, and in what situation Kruskal-Wallis Test was used, etc.
  3. In the discussion, authors have compared the findings with other Western countries, but how about other Asian countries which are culturally more similar to Taiwan, for example Mainland China, Japan, Korean? Please supplement the comparable information.

Author Response

Dear reviewer:

Thanks for your kind suggestion. I have revised the manuscript and answered your question. I show the important modifications below.

Question 1:

Nearly 70% of studies on dementia prevalence in long-term care facilities have been done in Europe.[16] Only limited studies have been conducted in Asia, the Americas, Africa or Eurasia. (Appendix C). Chen et al[17] in 2007 reported that 48% of residents of long-term care wards in Taiwan had dementia. Guo[18] and Xu[19] et al subsequently reported 36.7% and 44.5% in China. Therefore, a study in Asia that comprehensively covers all institutions and minimizes sampling bias is needed to provide a holistic perspective on the topic.

The prevalence of dementia in long-term care institutions would be dynamic[22]and up to date research is crucial for public health policy. Due to the rapid aging of society in recent years and the massive increase in the population of dementia, the National Health Re-search Institutes in Taiwan conducted an epidemiological survey on the prevalence of dementia in long-term care facilities.

Question 2:

The Statistical Analysis paragraph is revised as follows.

The data were expressed as mean (standard deviation) and number (%) for continuous and categorical variables, respectively. The group differences results were examined using with the Kruskal-Wallis t Test and Chi-Square test for continuous and categorical variables, respectively. We assessed weighted prevalence of dementia by using SUDAAN software (version 11.0.1, RTI International, Research Triangle Park, NC, USA) to account for sampling effects. The rest of statistical analyses in this study were performed by SAS (version 9.4 for Windows; SAS Institute, Inc., Cary, NC, USA).

Question 3:

I have added Appendix C (Review of Prevalence of Dementia Prevalence in Long-Term Care Facilities), which lists the data by geographic location and year of publication.

Finally, we would like to share the prevalence of dementia in Taiwan's long-term care facilities and call on the society to give more attention and resources to the "silent epidemic" disease that is dementia.

Reviewer 2 Report

Thank you very much for allowing me to review this manuscript.

I find the topic very interesting due to the progressive aging of the population worldwide.  The data resulting from this research are quite useful and have a great implication on care practice and on decision making at the political and organizational level.

I liked the manuscript very much and it is quite clear. It seems to me a very timely and necessary study for society to wake up and see the "silent epidemic" that is dementia.  Knowing the current data about it and the present and future consequences it has due to its prevalence.

The authors have conducted an epidemiological survey on the prevalence of dementia in long-term care facilities in Taiwan just before the outbreak of the Covid-19 pandemic. The objectives of this study were to realize the latest actual situation, the need for institutional care for the elderly with dementia and reference basis for government agencies to formulate dementia-related care policies.

The proposed aims seem realistic to me and the results provide useful information for decision-making. In addition, the authors report a high response rate that supports the results.

The Introduction section is adequate to put the readers in situation. Although it uses some references (numbers 6, 11, 16 and 17) that are very old. It would be convenient to update them.

The materials and methods section needs some improvements, which I will explain below:

- It would be appropriate to begin this section by mentioning the type and design of the study. It would also indicate the period of data collection and the place (institutions and country). To continue by explaining the various sections included in this section (Long-Term Care Facilities, Estimated of subjects, Sampling method: Two-stage random sampling, Statistical Analysis).

- This section needs further clarification on the variables that have been analyzed. Indicate which variables were analyzed and how they were collected. For example, if variables such as activities of daily living were collected, were scales such as Barthel's used?. Or what instruments/scales of measurement have been used, ad hoc questionnaires, Lawton and Brody...?.  Because in the results section these variables are mentioned and there is no information about it in the methods section. So we do not know which variables have been collected, nor the instrument used for their collection.

- The statistical analysis performed and the handling of the variables analyzed also need further explanation. Indicate why these statistical tests were performed and not others.

- Information on the approval of the ethics committee would be advisable to appear at the end of the section on materials and methods.

In the results section:

- In subsection 3.3. To indicate that the information discussed in the first two sentences is presented in Table 1.

- In the tables there are acronyms that are not explained at the foot of the table (e.g., table 1. ADL, IADL...). Review all tables.

- Even in one of the headings there are acronyms (ADL, IADL) that must be written in full the first time they appear. Include them in the text.

- In Table 2, revise the format so that the width of the "Normal elderly" column (N) is the same as the rest of the columns.

The discussion section seems appropriate and clearly expressed, except for the sentence: “In addition, three counties: Miaoli County, Yilan County, and Lianjiang County have been added.” This idea is not clear to people who do not know the organization of your country.

Finally, it would be necessary to include a paragraph containing the final conclusions of this study in response to its aims.

Author Response

Dear reviewer:

Thank you for your kind comments and great suggestions. As a clinical neurologist, I’m really touched by your "silent epidemic" statement.

Based on your suggestion, I have made detailed revisions to the manuscript.

-I have added Appendix C (Review of Prevalence of Dementia Prevalence in Long-Term Care Facilities), which lists more timely and representative references by geographic location and year of publication.

-The materials and methods section was revised and began this section by mentioning the type and design of the study.

-Variables including ADL and IDAL were explained.

- The Statistical Analysis paragraph is revised as follows.

The data were expressed as mean (standard deviation) and number (%) for continuous and categorical variables, respectively. The group differences results were examined using with the Kruskal-Wallis t Test and Chi-Square test for continuous and categorical variables, respectively. We assessed weighted prevalence of dementia by using SUDAAN software (version 11.0.1, RTI International, Research Triangle Park, NC, USA) to account for sampling effects. The rest of statistical analyses in this study were performed by SAS (version 9.4 for Windows; SAS Institute, Inc., Cary, NC, USA).

This study was reviewed and approved by the Medical Research Ethics Committee of the National Health Research Institutes, number EC1080502. All subjects or their family members signed an informed consent form.

In the results section:

- In subsection 3.3. More detail information was explained. (Table2)

- All tables were reviewed and acronyms that are explained at the foot of the table.

- Acronyms (ADL, IADL) are written in full the first time they appear.

- In Table 2, I revise the format so that the width of the "Normal elderly" column (N) is the same as the rest of the columns.

  • Discussion:

“In addition, three counties: Miaoli County, Yilan County, and Lianjiang County have been added.”  This sentence was re-write as below.

In addition, there are three more counties than the previous study [17] including Miaoli County, Yilan County and Lianjiang County.

-Conclusion:

In conclusion, we show that in a rapidly aged society, the prevalence of all-cause dementia in long-term care institutions can be as high as 87.1%. The dynamics of dementia prevalence in long-term care units reminds the importance of timely health policy and social resources. This study was exactly completed before outbreak of COVID-19 in Taiwan and could provide precious hallmark for future epidemiological research.

Finally, we would like to share the prevalence of dementia in Taiwan's long-term care facilities and call on the society to give more attention and resources to the "silent epidemic" disease.

This manuscript is a resubmission of an earlier submission. The following is a list of the peer review reports and author responses from that submission.